# MASDiff: Large Scale Multi-Agent System Emergence Control via Evolutionary Diffusion

## Abstract

Reward model learning methods are primarily divided into implicit reward modeling(IRM) and explicit reward modeling. Implicit reward modeling aims to learn the intrinsic reward of each agent to facilitate better explorationwhile explicit reward modeling(ERM) aims to learn the behavioral preferences of agents. The biggest difference between implicit and explicit reward modeling is that ERM is transferable to other scenarios but IRM cannot. Currently, few methods can simultaneously archive the global target and also learn the ERM. However, the problem addressed in this paper requires the integration of the two objectives. This paper use the diffusion model to generate the expert data for learning ERM. Since the traditional diffusion model can only generate data according to the given expert data, we introduce evolutionary diffusion model to generate data in the absence of any expert data. To steer the collaboration of all agents towards the specified macro-level objective, the macro-level objective is adopted as the fitness for the population. This mechanism transforms the multi-agent exploration based on intrinsic reward into the evolutionary exploration based on genetic operators. Moreover, the optimal individual retention strategy in the evolutionary diffusion model can address the non-stationary problem in MARL. In the experiments, we demonstrate that MASDiff can simultaneously archive the two objectives. Furthermore, we demonstrate that the ability to conduct counterfactual reasoning with the transferable ERM in different scenarios. We propose several 'What if' questions to indicate the change of scenarios and obtain relatively accurate counterfactual reasoning results.

## 1 Introduction

Reward model learning methods are primarily divided into implicit reward modeling and explicit reward modeling. Implicit reward modeling aims to learn the intrinsic reward of each agent to facilitate better exploration, ultimately achieving the overall outcomes. Intrinsic reward shaping(Cao et al., 2023), self-supervised RL(Pecháč et al., 2024) and credit assignment(Zhou et al., 2020) can be classified as implicit reward modeling. Explicit reward modeling aims to learn the behavioral preferences of agents by explicitly learning a reward model. The purpose is to ensure that the distribution of behavioral patterns derived from the reward model aligns with the statistical distribution observed in the expert data. Inverse RL(Wang et al., 2024), behavior cloning(Zhou et al., 2024) and Reinforcement Learning with Human Feedback(RLHF)(Casper et al., 2023) can be classified as implicit reward modeling. Currently, there is hardly any method capable of simultaneously achieving the two aforementioned objectives, namely, learning explicit reward models to realize macro objectives in multi-agent systems. This is primarily because existing approaches are mainly designed to address conventional reinforcement learning problems, which do not require the integration of these two objectives. However, the problem addressed in this paper requires the integration of the two objectives, i.e. to learn the explicit reward model for each agent on the one hand and to archive the overall outcome. We first present the specific problem to be solved in this study. Then, we give the contributions of this paper.

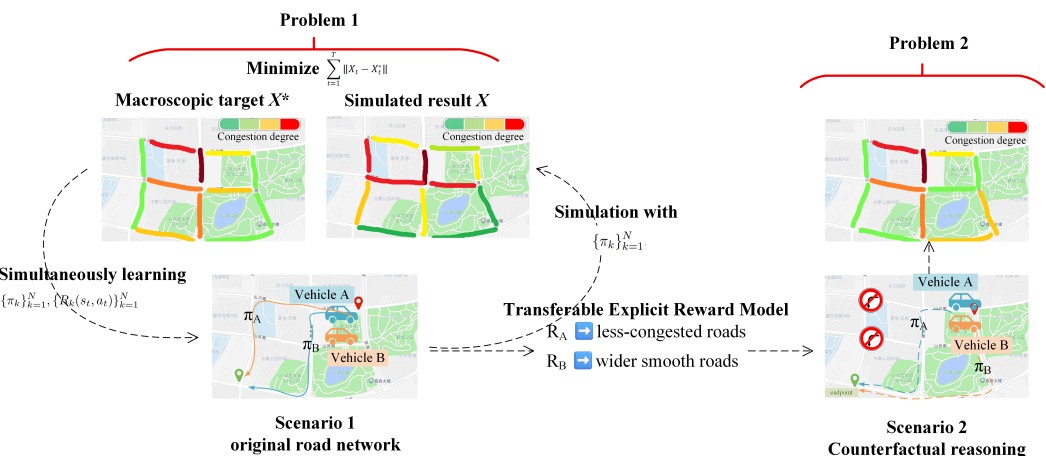

Figure 1: Example of the problems to be addressed by this paper

Consider the following two problems illustrated in Fig 1:

Problem 1: In scenario 1, can we learn the navigation policy of each vehicle such that the simulated congestion conditions aligns with the real congestion status? We assume that the number of vehicles is known and the origin-destination pairs for each vehicle are given.

Problem 2: In scenario 2, what the congestion conditions will be if we prohibit the left turns at two intersections?

If only problem 1 needs to be solved, we can use the implicit reward modeling approach in MARL. This method can learn the navigation policy instead of the explicit reward model for each vehicle to reconstruct the real congestion condition. But in order to solve problem 2 which is more important, the explicit reward model for each vehicle also need to be learned when solving problem 1. Because although the vehicle's optimal route may be different in various scenarios, its decision preference remains the same. For example, vehicle A prefers less-congested roads, while vehicle B prefers wider arterial roads. In scenario 1, vehicle As route is shown as the blue line and vehicle Bs as the orange line. In scenario 2, vehicle A stays on the same route (blue dashed line), while vehicle B shifts to another road (orange dashed line). So to learn the explicit reward model, we need to obtain the expert data for each vehicle in scenario 1 which is transferable in scenario 2. To answer the what-if questions question 2, vehicles simply need to learn the new navigation policy based on the reward model as illustrated in Figure 1. In real world, many issues are similar to problem 1 and 2(Jiaoling et al., 2025). Problem 1 can be generalized as Formula 1.

$$\{\pi_k\}_{k=1}^N, \{R_k(s_t, a_t)\}_{k=1}^N = \text{Argmin} \sum_{t=1}^T \|X_t - X_t^*\|$$

$$\text{s.t.} \tag{1}$$

$$\pi_k = \arg\max \mathbb{E}\left[\sum_{t=1}^T \gamma^t r_{k,t}\right] \quad \text{and} \quad r_{k,t} = R_k(s_t, a_t)$$

We need to learn $\{\pi_k\}_{k=1}^N$ to cooperatively minimize $\sum_{t=1}^T \|X_t - X_t^*\|$. And also we need to learn $R_k(s_t, a_t)$ for so that $\pi_k$ can maximize $\sum_{t=1}^T \gamma^t r_{k,t}$. Traditional cooperative MARL methods such as MAPPO(Yu et al., 2022) or QMIX(Rashid et al., 2020) only require $\{\pi_k\}_{k=1}^N$ instead of $R_k(s_t, a_t)$ to be learned. Traditional explicit reward modeling methods such as GAIL(Ho & Ermon, 2016),RLHF(Christiano et al., 2017) only require $R_k(s_t, a_t)$ instead of $\{\pi_k\}_{k=1}^N$ to be learned. When the number of agents is 1000+, problem 1 becomes the emergence control problem. Through interactions of agents guided by microscopic policies, different types of macroscopic properties, such as different types of congestion patterns, may emerge. Although from a research perspective,

1000+ agents pose significant challenges for both algorithmic design and hardware implementation. From a practical perspective, it is highly prevalent. For example, regional traffic networks typically involves 1000+ vehicles. Airport terminals usually accommodate 1000+ pedestrians. Problem 2 is actually the counterfactual reasoning based on the explicit reward model obtained by solving problem 1. To distinguish from the problems typically addressed by existing MARL, we collectively refer to the solutions for problem 1 and 2 as emergence control.

The solution to this problem primarily involves two main challenges: (1)To maximize $\sum_{t=1}^{T} \|X_t - X_t^*\|$ aligns with the goal of implicit reward modeling(Kontogiannis et al., 2025). To explicitly learn the navigation preference $R_k(s_t, a_t)$ for each vehicle $k$ aligns with the goal of explicit reward modeling. However, a simple combination of these two methods cannot effectively solve the integrated problem posed by their superposition. (2) Currently, the primary computational frameworks capable of supporting large-scale multi-agent reinforcement learning (involving 1000+ agents) are population-based MARL and Swarm RL. However, population-based MARL operate on the parameters of policies(Li et al., 2023). It cannot learn an explicit reward model for each individual agent. Swarm RL such as the Mean Field Reinforcement Learning requires the agents in the swarm are interchangeable. Neither of these frameworks can solve the emergence control problem.

We propose the MASDiff training framework that utilizes an evolutionary diffusion model. The evolutionary diffusion model can simultaneously learn the explicit reward model for 1000+ agents and enhance the cooperation of multi-agents to reach the macroscopic target. The contributions of this paper are: **(1)**Since we don't have the expert dataset for training explicit reward model. We use the diffusion model to generate the expert data. **(2)**Since the traditional diffusion model can only generate data according to the given data, we introduce evolutionary diffusion model to generate data in an evolutionary manner. This mechanism dose not require any training data and guarantees that we can obtain the explicit reward model $\{R_k(s_t, a_t)\}_{k=1}^{N}$. **(3)**To steer the collaboration of all agents towards the specified macro-level objective, MASDiff employs the macro-level objective, i.e. $\text{Argmin} \sum_{t=1}^{T} \|X_t - X_t^*\|$ in Formula 1, as the fitness for the evolutionary population. It transforms the multi-agent exploration based on intrinsic reward into the evolutionary exploration based on genetic operators. **(4)**The optimal individual retention strategy in the evolutionary diffusion model can address the non-stationary problem in MARL. **(5)**MASDiff adopt the DTDE and offline-to-online-to-offline training framework to support MARL with 1000+ agents. Since the evolutionary diffusion model can generate the training data for each agent, the learning process of each agent can be executed in parallel by using offline RL for each agent. Training of instance in population can be executed in parallel in an offline manner. Only sample collection are conducted online.

## 2 RELATED WORK

**Explicit reward modeling.** Current approaches for explicit reward modeling fall into three main categories. **(1)Behavioral Cloning (BC).** BC method learns intelligent body actions directly (Hussein et al., 2017)(Le Mero et al., 2022)(Zhou et al., 2024). It does not require learning reward functions but needs a large amount of expert trajectory data, split into state-policy pairs to represent the experts actions in the current state. **(2)Inverse Reinforcement Learning (IRL).** IRL infers a reward function by modeling expert trajectories, and learning a reward function that emulates expert decision patterns(Adams et al., 2022)(Fernando et al., 2021)(Wang et al., 2024). **(3)Reinforcement Learning with Human Feedback (RLHF).** RLHF aligns agents preferences with real-world behavior by using human feedback to optimize reward functions(Casper et al., 2023)(Kaufmann et al., 2025)(Cao et al., 2023). All three methods require large-scale trajectory datasets, which is problematic in our case due to the lack of granular trajectory data for individual vehicles.

**Implicit Reward modeling.** Implicit reward modeling aims to learn the intrinsic reward of an agent to facilitate better exploration in multi-agent reinforcement learning, ultimately achieving superior overall outcomes. **(1)Single-Agent Exploration.** Reward bonuses based on the inverse state-action count have been shown to be effective in accelerating learning (Strehl & Littman, 2008). In order to scale count-based approaches to large state spaces, state counts are use as reward bonuses (Ostrovski et al., 2017)(Tang et al., 2017). Some work has focused on defining intrinsic rewards for exploration based on inspiration from psychology (Oudeyer & Kaplan, 2007). **(2)Multi-Agent Exploration.** MAVEN (Mahajan et al., 2019) adopts hierarchical control and the policies of the agents are conditioned on the shared latent variable generated by a hierarchical policy. Wang et al. (2020)

propose two exploration methods, EITI and EDTI, to induce cooperative exploration by capturing the influence of one agents on other agents. CMAE (Liu et al., 2021) proposes that reward function only depends on a small subset of the large state space. EMC (Zheng et al., 2021) proposes that local Q function of each agent can capture the novelty of states and the influences between agents.

Both of the explicit and implicit reward model learning methods are unable to solve our problem. The explanation are given in the above paragraphs.

**MARL training architecture for 1000+ agents.** Currently, the primary computational frameworks capable of supporting large-scale multi-agent reinforcement learning are population-based MARL and Swarm MARL. **(1)Population-based MARL.** Population-based MARL is a feasible approach to support distributed learning. Prior works include population-based training (Carroll et al., 2019)(Jaderberg et al., 2019), self-play (Vinyals et al., 2019)(Heinrich et al., 2015) and meta-game (Muller et al., 2020)(Omidshafiei et al., 2019). However, population-based MARL cannot learn an explicit reward model for each individual agent. **(2)Swarm MARL.** MARL for swarm system is described as a swarm MDP environment (Hüttenrauch et al., 2019). Among the MARL methods for swarm systems, mean field theory are wildly adopted. Mean Field Reinforcement Learning is proposed to solve the scalability and the interaction of the population of agents(Guo et al., 2023)(Gu et al., 2023)(Anahtarci et al., 2023)(Perrin et al., 2020). However, those methods require two important properties of swarm systems. (1) the agents in the swarm are interchangeable.(2) the exact number of agents in the swarm is irrelevant. It also cannot learn an explicit reward model for each individual agent.

**Diffusion models for reinforcement learning.** Offline reinforcement learning is a powerful tool for addressing the problem of action sequence combination explosion. However, its performance is ultimately constrained by the offline dataset. Many studies have introduced diffusion models into reinforcement learning(Kang et al., 2023),(Hansen-Estruch et al., 2023),(Ren et al., 2025) . Zhu et al. (2023) summarizes the application of diffusion models in reinforcement learning-related fields. Janner et al. (2022) adopts diffusion models as planners to replace traditional autoregressive-based planners. SafeDiffuser (Xiao et al., 2025) introduces safety constraints to generate safe trajectories. Diffusion-QL (Wang et al., 2023) integrates diffusion policies into the Q-learning framework to address the distribution shift issue in offline RL. CEP (Lu et al., 2023) uses contrastive performance prediction to guide energy function sampling. MADiff (Zhu et al., 2024) uses an attention-based diffusion model to model the complex coordination among behaviors of multiple agents. All those methods require training data and thus is not applicable for solvin our problem.

## 3 METHODOLOGY

### 3.1 IDEAL TRAINING FRAMEWORK VIA ORACLE

We define a Markov Decision Process as $(S, A, R, \boldsymbol{\pi})$, where $S = \{s_1, s_2, ..., s_N\}$, $A = \{a_1, a_2, ..., a_N\}$ and $R = \{r_1, r_2, ..., r_N\}$ are the joint set of individual states, actions and rewards, respectively. Once an agent reaches its next state, it receives a reward $r$. $\boldsymbol{\pi} = \{\pi_i\}_{i=1}^N$ are the combination of policies. Let $\tau_{\pi_i} = \{(s, a, s')_t\}_{t=1}^T$ denote trajectory sampled by $\pi_i$ where $s, s' \in S_i$ and $a \in A_i$. Let $\boldsymbol{\tau} = \{\tau_{\pi_i}\}_{i=1}^N, \boldsymbol{\tau} \in \mathbb{R}^{|S|*|a|*|S|*|T|*|N|}$ denote the combined trajectory collected by $\{\pi_i\}_{i=1}^N$. Let $R = \{\boldsymbol{r}_i\}_{i=1}^N, \boldsymbol{r}_i = \{r_t\}_{t=1}^T$, $\boldsymbol{r}_i$ denotes the corresponding rewards for $\tau_{\pi_i} \in \boldsymbol{\tau}$.

Each $\pi_i \in \boldsymbol{\pi}$ can be obtained by conducting offline reinforcement learning on $\{\tau_{\pi_i} \cup \boldsymbol{r}_i\} \in \{\boldsymbol{\tau} \cup R\}$ via the DTDE framework. Let $\{\{X_t\}_{t=1}^T\}_{\{\boldsymbol{\tau} \cup R\}}$ denotes the macroscopic status by conducting simulation on $\boldsymbol{\pi}$ deriving from $\{\boldsymbol{\tau} \cup R\}$. Let $\{X_t^*\}_{t=1}^T$ denotes the target macroscopic state. If we can obtain the optimal $\{\boldsymbol{\tau} \cup R\}$, then the optimal $\boldsymbol{\pi}$ can be derived. This paper firstly designs an ideal training framework based on an oracle $\mathcal{O}$ to obtain the optimal $\{\boldsymbol{\tau} \cup R\}$.

**Oracle reward model.** We denote by $\mathcal{O}(\boldsymbol{\tau})$ an oracle reward model satisfying the following guarantee for $\boldsymbol{\tau}$. For input $\boldsymbol{\tau}$, the oracle outputs $R \leftarrow \mathcal{O}(\boldsymbol{\tau})$. For any other $R'$, $R' \neq R$, we have $\sum_{t=1}^T ||X_t - X_t^*|| \leq \sum_{t=1}^T ||X_t' - X_t^*||$, where $X_t \in \{\{X_t\}_{t=1}^T\}_{\{\boldsymbol{\tau} \cup R\}}$ and $X'_t \in \{\{X_t'\}_{t=1}^T\}_{\{\boldsymbol{\tau} \cup R'\}}$.

**Proposition 1.** $\sum_{t=1}^T ||X_t - X_t^*||$ is monotonically decreasing in each iteration of Algorithm 1.

---

**Algorithm 1** Ideal training framework

---

1: Input: Oracle reward model $\mathcal{O}$, target macro emergent phenomenon $\{X_t\}_{t=1}^T$
2: Initializing policy combination $\boldsymbol{\pi} = \{\pi_i\}_{i=1}^N$
3: **while** $\sum_{t=1}^T \|X_t - X_t^*\|$ not converged **do**
4:    Conduct simulation with $\boldsymbol{\pi} = \{\pi_i\}_{i=1}^N$, obtaining $\boldsymbol{\tau}$ and $\{X_t\}_{t=1}^T$
5:    Query Oracle $\mathcal{O}$ to obtain $R$, $R \leftarrow \mathcal{O}(\boldsymbol{\tau})$
6:    Update $\boldsymbol{\pi} = \{\pi_i\}_{i=1}^N$ by offline RL via DTDE framework with $\{\boldsymbol{\tau} \cup R\}$
7: **end while**

---

**Proof.** Based on oracle reward model $\mathcal{O}$, the MARL in Algorithm 1 is actually transformed into a single-agent reinforcement learning. $\{X_i\}_{i=1}^T$ can be viewed as the ordinary reward value. The entire process is equivalent to classical policy iteration, which has been proven to yield monotonic improvement in returns, corresponding to a monotonic decrease of $\{X_i\}_{i=1}^T$ in Algorithm 1.

Based on Proposition 1, we introduce the MASDiff framework that use a diffusion model to approximate oracle $\mathcal{O}$. MASDiff shift the focus from exploring the optimal $\boldsymbol{\pi}$ to exploring the optimal $R$ on condition of $\boldsymbol{\tau}$. In the article "Reward is enough" (Silver et al., 2021), the author believes rewards are sufficient to drive behaviors. Therefore, the diffusion model essentially generates the decision preferences for agents.

## 3.2 APPROXIMATING ORACLE $\mathcal{O}$ WITH DIFFUSION MODEL

To approximate oracle $\mathcal{O}$, the diffusion model is to generate $R$ on condition of $\boldsymbol{\tau}$. From a practical perspective, it is also necessary to treat $\boldsymbol{\tau}$ as a condition obtained from the environment. Because the generated $\boldsymbol{\tau}$ may be inconsistent with reality. This paper adopts the denoising diffusion probability model (Ho et al., 2020) as the basic diffusion model. We firstly give the description of the basic diffusion model. Then, we describe how to train the model such that it can approximate $\mathcal{O}$.

### 3.2.1 CONDITIONAL DENOISING DIFFUSION PROBABILITY MODEL

To generate $R$ by incorporating $\boldsymbol{\tau}$ as conditions, the original distribution $R_0$ of the diffusion process can be transformed into an isotropic Gaussian distribution $\mathcal{N}(0,1)$ through $T$ steps of forward diffusion. The diffusion model denoted as $p_\theta$ attempts to recover the original distribution of $R_0$ under the condition of $\boldsymbol{\tau}$. Moreover, the dimension of $\boldsymbol{\tau}$ is $\mathbb{R}^{|S|*|a|*|S|*|T|*|N|}$ which is quite high when $N>1000$ and $T>100$. We reduce the dimensions of $\boldsymbol{\tau}$ and $R$ with encoders by mapping $\boldsymbol{\tau}$ and $R$ to latent space representation $Z_{\boldsymbol{\tau}}$ and $Z_R$.

The forward process is parameterized as:

$$q\left(Z_{R1:T} \mid Z_{R0}, Z_{\boldsymbol{\tau}}\right) = \prod_{t=1}^T q\left(Z_{Rt} \mid Z_{Rt-1}, Z_{\boldsymbol{\tau}}\right) \tag{2}$$

$$q\left(Z_{Rt} \mid Z_{Rt-1}, Z_{\boldsymbol{\tau}}\right) = \mathcal{N}\left(Z_{Rt}; \sqrt{1-\beta_t}Z_{Rt-1}, \beta_t I\right) \tag{3}$$

where $q\left(Z_{Rt} \mid Z_{Rt-1}, Z_{\boldsymbol{\tau}}\right)$ represents the conditional Gaussian distribution, and the variance $\beta_t$ adjusts the noise level. $Z_{Rt}$ at any time step $t$ can be directly represented by $Z_{R0}$ as:

$$Z_{Rt} = \sqrt{\alpha_t}Z_{R0} + \sqrt{1-\alpha_t}\epsilon, \quad \epsilon \sim \mathcal{N}(0,1) \tag{4}$$

$$\alpha_t = \prod_{s=1}^T (1-\beta_s) \tag{5}$$

The backward process is parameterized as:

$$p_\theta\left(Z_{R0}, \ldots, Z_{Rt-1} \mid Z_{RT}, Z_{\boldsymbol{\tau}}\right) = p_\theta\left(Z_{RT}\right) \prod_{t=1}^T p_\theta\left(Z_{Rt-1} \mid Z_{Rt}, Z_{\boldsymbol{\tau}}\right) \tag{6}$$

where $Z_{RT} \sim \mathcal{N}(0,1)$ and $p_\theta\left(Z_{Rt-1} \mid Z_{Rt}, Z_{\boldsymbol{\tau}}\right)$ is assumed to follow a normal distribution with learnable parameters. The backward process can be trained by optimizing the following objective:

$$L = \min_\theta \mathbb{E}_{Z_{Rt}, Z_{\boldsymbol{\tau}}, \epsilon, t}\left[\|\epsilon - \epsilon_\theta\left(Z_{Rt}, Z_{\boldsymbol{\tau}}, t\right)\|_2^2\right] \tag{7}$$

where $\epsilon_\theta\left(Z_{Rt}, Z_{\boldsymbol{\tau}}, t\right)$ predicts the noise using a neural network.

### 3.2.2 TRAINING OF DIFFUSION MODEL

Training of diffusion model implies two layers of meaning. Firstly, let $(\{\boldsymbol{\tau} \cup R\}, \rho(\{\boldsymbol{\tau} \cup R\}))$ denote a training sample for $p_\theta$. $\rho(\{\boldsymbol{\tau} \cup R\}) = \sum_{t=1}^{T}\|X_t - X_t^*\|$ denotes the weight of $\{\boldsymbol{\tau} \cup R\}$, where $X_t \in \{\{X_t\}_{t=1}^{T}\}_{\{\boldsymbol{\tau} \cup R\}}$. $\rho(\{\boldsymbol{\tau} \cup R\})$ is normalized based on Formula 8. $\min_\rho$ and $\max_\rho$ denotes the minimum and maximum value of $\rho$ in the sample set. To obtain high-quality samples, $\gamma$ is further applied for scaling. If we have enough samples, we can train the model according to Formula 9 (Kumar & Levine, 2020). The weighted training can let the diffusion model generate sample with higher $\rho$ values.

$$\rho(\{\boldsymbol{\tau} \cup R\}) = \left(1 - \frac{\rho - \min_\rho}{\max_\rho - \min_\rho}\right) * \gamma \tag{8}$$

$$L = \min_\theta \mathbb{E}_{R_t, \boldsymbol{\tau}, \epsilon, t}\left[\rho(\{\boldsymbol{\tau} \cup R\}) * \|\epsilon - \epsilon_\theta\left(R_t, \boldsymbol{\tau}, t\right)\|_2^2\right] \tag{9}$$

Secondly, since there is not any samples at the beginning, we need to train $p_\theta$ in a bootstrapping manner. In other words, $p_\theta$ need to generate training sample for itself. Moreover, $p_\theta$ need to explore samples that have higher $\rho$ value. We design the evolution mechanism to let $p_\theta$ explore better new samples. The evolutionary strategy consists of four steps illustrated as follows.

**Population.** Let $\mathcal{T} = \{\{\boldsymbol{\tau} \cup R\}_j\}_{j=1}^{M}$ denote the population of $\{\boldsymbol{\tau} \cup R\}$. $\mathcal{T}$ consists $M$ instances of $\{\boldsymbol{\tau} \cup R\}$ where $R \sim p_\theta(R|\boldsymbol{\tau})$. Let $\Pi = \{\boldsymbol{\pi}_j\}_{j=1}^{M}$, $\boldsymbol{\pi} = \{\pi_i\}_{i=1}^{N}$ denote the policy population. $\Pi$ consists $M$ instances of $\boldsymbol{\pi}$. We randomly initialize the policy population $\Pi$. The initial population of $\mathcal{T}$ is the rollout data by $\Pi$.

**Sample scoring.** At each iteration $k$, we score each $\{\boldsymbol{\tau} \cup R\} \in \mathcal{T}^k$ with $\rho(\{\boldsymbol{\tau} \cup R\})$.

**Selection.** We use $\rho(\{\boldsymbol{\tau} \cup R\})$ to decide which samples in $\mathcal{T}^k$ should be selected to propagate to the next iteration. Let $E^k$ denotes the elite set selected from $\mathcal{T}^k$ according to the roulettewheel strategy (Lipowska, 2012).

**Mutation using truncated diffusion-denoising.** We apply randomized mutations to $E^k$ for exploration. A truncated diffusion-denoising process is adopted to mutate trajectories. We run the first $t$ steps of the forward diffusion process to add noise to elite trajectories in $E^k$ based on Formula 10.

$$E_{noisy}^{k+1} = \{\boldsymbol{\tau} \cup R_{noisy}\}, R_{noisy} = \{\sqrt{\alpha_t}R + \sqrt{1-\alpha_t}\epsilon, \epsilon \sim N(0,1)|\{\boldsymbol{\tau} \cup R\} \in E^k\} \tag{10}$$

Then we run the last $t$ steps of the reverse diffusion process to denoise $E_{noisy}^{k+1}$ to obtain clean $E_{clean}^{k+1}$ based on Formula 11. Finally, $E_{clean}^{k+1}$ is the next generation of $\mathcal{T}^k$, i.e. $\mathcal{T}^{k+1}$.

$$E_{clean}^{k+1} = \{\boldsymbol{\tau} \cup R_{clean}\}, R_{clean} \sim \{p_\theta(R_{noisy}|\boldsymbol{\tau})|\{\boldsymbol{\tau} \cup R_{noisy}\} \in E_{noisy}^{k+1}\} \tag{11}$$

**Notation.** Both the diffusion model and $\{X_t\}_{t=1}^{T}$ incorporate $T$. $T$ denotes the diffusion steps when describing the diffusion model, whereas $T$ represents the time-series length when referring to $\{X_t\}_{t=1}^{T}$. In Formula 10 and 11, we use $R$ and $\boldsymbol{\tau}$ to denote $Z_R$ and $Z_{\boldsymbol{\tau}}$ for brevity.

## 3.3 TRAINING FRAMEWORK OF MASDIFF TO SUPPORT MARL WITH 1000+ AGENTS

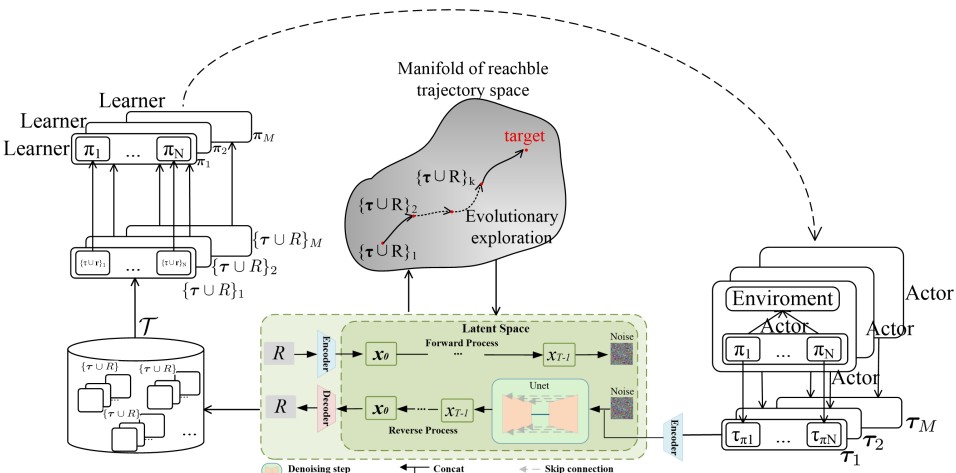

Figure 2: Training framework of MASDiff

**Training framework of MASDiff.** Algorithm 2 illustrates how to transform the ideal training framework into the practically feasible training framework MASDiff. MASDiff utilizes policy population and sample population to leverage evolutionary search over the trajectory manifold.

---

**Algorithm 2** Training framework of MASDiff

---

1: Input:Target macro emergent phenomenon $\{X_t\}_{t=1}^T$. number of agents $N$
2: Initializing policy population $\Pi,\Pi = \{\boldsymbol{\pi}_j\}_{j=1}^M$ ,$\boldsymbol{\pi} = \{\pi_i\}_{i=1}^N$
3: Initializing diffusion model $p_\theta$
4: **while** $\sum_{t=1}^T ||X_t - X_t^*||$ not converged or max iteration not reached **do**
5:     Simulating with $\Pi = \{\boldsymbol{\pi}_j\}_{j=1}^M$ , obtaining $\{\boldsymbol{\tau}\}_{j=1}^M$ for each $\boldsymbol{\pi}_j$   ▷ collect trajectories online
6:     Query $p_\theta$ to obtain $R_j$ for each $\boldsymbol{\tau}_j \in \{\boldsymbol{\tau}\}_{j=1}^M$ , $R_j \sim p_\theta(R_j|\boldsymbol{\tau}_j)$
7:     Update each $\boldsymbol{\pi}_j \in \Pi$ by independent offline RL with $\{\boldsymbol{\tau}_j \cup R_j\}$     ▷ update $\boldsymbol{\pi}_j \in \Pi$ offline
8:     Simulating with $\Pi = \{\boldsymbol{\pi}_j\}_{j=1}^M$ , obtaining $\{\{X_t\}_{t=1}^T\}_j\}_{j=1}^M$ for each $\boldsymbol{\pi}_j$ ▷ collect simulation result for trajectories in line 5 online
9:     Update $p_\theta$ with weighted sample $\{\{\boldsymbol{\tau}_j \cup R_j\}, \rho(\{\boldsymbol{\tau}_j \cup R_j\})\}_{j=1}^M$
10:     Evolving the sample population $\{\boldsymbol{\tau}_j \cup R_j\}_{j=1}^M$ to the next generation $\{\boldsymbol{\tau}_j \cup R'_j\}_{j=1}^M$
11:     Update each $\boldsymbol{\pi}_j \in \Pi$ by independent offline RL with $\{\boldsymbol{\tau}_j \cup R'_j\}$   ▷ update $\boldsymbol{\pi}_j \in \Pi$ offline
12:     Simulating with $\Pi = \{\boldsymbol{\pi}_j\}_{j=1}^M$ , obtaining $\{\{X_t\}_{t=1}^T\}_j\}_{j=1}^M$ for each $\boldsymbol{\pi}_j$ ▷ collect simulation result for trajectories in line 10 online
13:     Update $p_\theta$ with weighted sample $\{\{\boldsymbol{\tau}_j \cup R'_j\}, \rho(\{\boldsymbol{\tau}_j \cup R'_j\})\}_{j=1}^M$
14: **end while**

---

In line 5, we use the policy population to collect trajectories. In line 6, rewards are generated by query the diffusion model with the trajectories. In line 7∼9, we update the diffusion model by samples collected in line 5∼6. In line 10∼13, the diffusion model are further updated by samples collected with evolutionary strategy. MASDiff also adopt the offline-to-online-to-offline training framework. Training are all conducted offline. Only sample collection and simulation result collection are conducted online.

**Recovering explicit reward model** $R_k(s_t, a_t)$ **from weighted sample** $\{\{\boldsymbol{\tau}_j \cup R'_j\}, \rho(\{\boldsymbol{\tau}_j \cup R'_j\})\}_{j=1}^M$**.** Assume $\{\{\boldsymbol{\tau}_j \cup R'_j\}, \rho(\{\boldsymbol{\tau}_j \cup R'_j\})\}$ has the highest $\rho$ value. Assume $\{\tau_{\pi_i} \cup \boldsymbol{r}_k\} \in \{\{\boldsymbol{\tau}_j \cup R'_j\}, \tau_{\pi_k} = \{(s,a,s')_t\}_{t=1}^T$ and $\boldsymbol{r}_k = \{r_t\}_{t=1}^T$. We can parameterize $R_k(s_t, a_t)$ using a single-time-step neural network $r_t = g_\theta(s_t, a_t)$ to predict a reward for a state-action pair $(s_t, a_t)$.

**Parallel training paradigm for MASDiff.** Although the dimension of the trajectory space is $\mathbb{R}^{|S|*|a|*|S|*|T|*|N|}$ which is quite large. The reachable trajectory space is a low dimensional manifold embedded in $\mathbb{R}^{|S|*|a|*|S|*|T|*|N|}$. MASDiff is essentially to efficiently explore the reachable trajectory manifold that can emerge the target macroscopic phenomenon. Figure 2 illustrates the training framework of MASDiff. It decouples the training and rollout tasks using the Actor-Diffusion-Learner model. The Learner operates policy training and the Actor operates data collecting. The diffusion model interleaves the Learners and Actors by providing better samples to the Learner instead of those collected directly by Actors.

To conduct fully decentralized multi-agent learning, MASDiff uses $\tau_{\pi_i} \in \boldsymbol{\tau}$ to train $\pi_i \in \boldsymbol{\pi}$ independently. One learner for one policy, policy learning is independent to other agents. Actor is to collect the rollout data $\{\tau_{\pi_i}\}_{i=1}^N$ using $\{\pi_i\}_{i=1}^N$. Multiple Actors can be dispatched in parallel. The Actor-Diffusion-Learner architecture are built on top of Ray which allows tasks to be distributed over a large cluster.

## 4 EXPERIMENT

The following three research questions illustrate the purpose of the experiment:

**RQ1:** Can MASDiff minimize $\sum_{t=1}^T \|X_t - X_t^*\|$?

**RQ2:** Can the explicit reward model $R_k(s_t, a_t)$ learned by MASDiff answer the 'What if' counterfactual questions in new scenarios?

**RQ3:** Since the search space of multi-agent reinforcement learning grows exponentially as the number of agents increases, how is the scalability of the MASDiff framework?

### 4.1 EXPERIMENTAL SETTINGS

**Comparison methods.** Since existing CTDE or DTDE MARL framework can not work on such large number of vehicles with local reward completely unknown. We design methods for comparison by change or remove part of the algorithms in MASDiff framework.

CGAN replaces the conditional denoising diffusion probabilistic model in MASDiff with Conditional Generative Adversarial Network.

LDM_IMIT adopts imitation learning instead of offline DQN in learning vehicle navigation policy.

LDM_CFG use classifier free guidance in the conditional diffusion model. It incorporates $\rho(\{\tau \cup R\})$ as an additional condition for the diffusion model, i.e. $R \sim p_\theta(R|\boldsymbol{\tau}_j, \rho(\{\tau \cup R\}))$. Instead of utilizing evolutionary strategy to explore new sample, LDM_CFG uses the additional condition $\rho(\{\tau \cup R\})$ to explore new sample.

LDM_PCFG add a proxy in LDM_CFG. The proxy uses $\sum_{t=1}^T \|X_t - X_t^*\|$ to predict $\rho(\{\tau \cup R\})$. The condition $\rho(\{\tau \cup R\})$ in LDM_PCFG is given by the proxy in stead of the real value of $\rho(\{\tau \cup R\})$.

**Experimental environment.** Experiments were conducted on a high-performance server equipped with an AMD EPYC 9554 CPU, four NVIDIA RTX 4090 GPUs (24GB each), 256GB of DDR5 RAM, and a 2TB NVMe SSD. The operation system is Ubuntu 22.04 LTS. The software stack include Python 3.10, the SUMO(Simulation of Urban MObility) simulation environment, and the Ray framework for parallel execution. The population size in Algorithm 2 is 500. The traffic network in 3 which is part of a certain district of Chongqing. This region includes 114 roads and 45 intersections.

### 4.2 RQ1: CAN MASDIFF MINIMIZE $\sum_{t=1}^T \|X_t - X_t^*\|$?

**Experimental design.** This experiment is conducted in scenario 1 illustrated in Figure 3(b). Figure 3(a) is the original Baidu map. Figure 3(b) is the SUMO traffic road network derived from Figure 3(a). The surveillance camera data from 8:30 to 9:00 am on August 21, 2023 in this region are used for this experiment. There are totally 1557 vehicles in the camera data. We adopt the largest functional cluster of the traffic network ($LCC$) as the emergent macro traffic state denoted as $\{LCC_t^*\}_{t=1}^T$. Functional cluster is the connected roads with relatively high speed (Zeng et al.,

2020). We calculate $LCC$ using the velocity of the 1557 vehicles recorded by surveillance camera. $LCC$ is calculated every minute and thus $T = 30$ in $\{LCC_t^*\}_{t=1}^T$. MASDiff learns the navigation policy for each vehicle according to $\{LCC_t^*\}_{t=1}^{30}$. $\{LCC_t\}_{t=1}^{30}$ is obtained by conduct simulation with the learned policies.

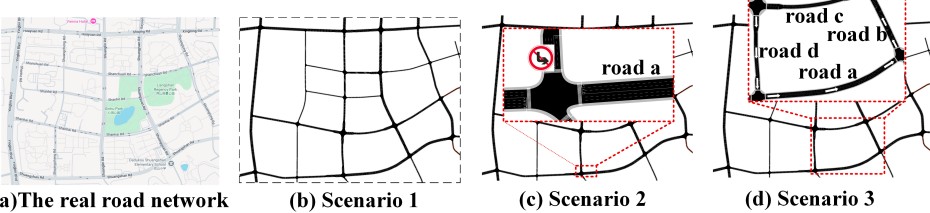

(a)The real road network    (b) Scenario 1    (c) Scenario 2    (d) Scenario 3

Figure 3: Different experiment scenarios

**Experimental results.** The results are shown in Figure 4. The horizontal axis represents the iteration times. The vertical axis represents the minimal value of $\|\{LCC_t\}_{t=1}^{30} - \{LCC_t^*\}_{t=1}^{30}\|$ during the past iterations. All methods are iterated 100 rounds. MASDiff obtain the best results. LDM_CFG and LDM_PCFG produce the worst results. This indicates that directly using $\sum_{t=1}^{T}\|X_t - X_t^*\|$ as a condition provides limited guidance for the diffusion model's exploration.

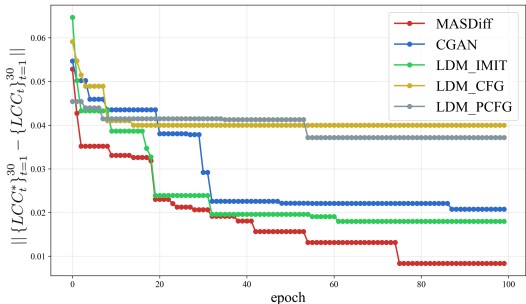

Figure 4: Minimal value of $\|\{LCC_t^*\}_{t=1}^{30} - \{LCC_t\}_{t=1}^{30}\|$ per iteration.

Figure 5 shows the detail of $\{LCC_t\}_{t=1}^{30}$ in each minute. The orange curve is $\{LCC_t^*\}_{t=1}^{30}$. The blue curve is $\{LCC_t\}_{t=1}^{30}$. The curve by MASDiff best approximates the ground truth.

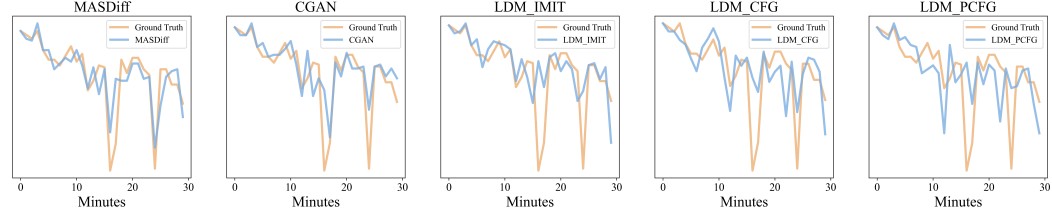

Figure 5: $\{LCC_t^*\}_{t=1}^{30}$ and $\{LCC_t\}_{t=1}^{30}$ in scenario 1.

### 4.3 RQ2: CAN WE CONDUCT COUNTERFACTUAL REASONING BY THE LEARNED EXPLICIT REWARD MODEL?

**Experimental design.** This experiment is conducted in scenario 2~3 illustrated in Figure 3(c)~(d).

Scenario 2: What is the $\{MQL_t\}_{t=1}^T$ if we prohibiting the left turns onto road a at the intersections?

Scenario 3: What is the $\{MQL_t\}_{t=1}^T$ if we establish a one-way traffic loop in the specific area?

$MQL$ is adopted as the macro emergent traffic state which is the maximum queue length of a road segment in a certain period. $\{MQL_t^*\}_{t=1}^{30}$ represents the ground truth data. $\{MQL_t^*\}_{t=1,s1}^{30} \sim \{MQL_t\}_{t=1,s3}^{30}$ denote the ground truth data in scenario $1 \sim 3$. $\{MQL_t\}_{t=1,s1}^{30} \sim \{MQL_t^*\}_{t=1,s3}^{30}$ denote the results obtained through different methods in scenario $1 \sim 3$.

Due to the change of scenario, $\{MQL_t^*\}_{t=1,s2}^{30}$ and $\{MQL_t^*\}_{t=1,s3}^{30}$ cannot be obtained from the original surveillance camera data or be predicted based on historical data. Therefore, we synthesize the ground truth data through randomly simulated reward models. First, we randomly generate the ground truth reward model for those 1557 vehicles. Based on the generated reward models, we conduct distributed RL for each vehicle in scenario $1 \sim 3$ to obtain the navigation policies. By simulating with the policies we can obtain $\{MQL_t^*\}_{t=1,s2}^{30}, \{MQL_t^*\}_{t=1,s3}^{30}$. To obtain $\{MQL_t\}_{t=1,s2}^{30}, \{MQL_t\}_{t=1,s3}^{30}$, We first learn the explicit reward model and the navigation policy according to $\{MQL_t^*\}_{t=1,s1}^{30}$. Then, $\{MQL_t\}_{t=1,s2}^{30}$ and $\{MQL_t\}_{t=1,s3}^{30}$ can be obtained by conduct simulation with the learned policies in scenario 2 and 3.

**Experimental results.** In scenario 1, roads a and b are congested. Roads c and d experience lower traffic volumes. In scenario 2, congestion on road a is partially alleviated. In scenario 3, congestion on both roads a and b is reduced. Figures 7, 8 and 9 show heatmaps of $\{MQL_t\}_{t=1,s1}^{30} \sim \{MQL_t\}_{t=1,s2}^{30}$ for each method in different scenarios. The horizontal axis represents the minutes. The vertical axis represents the $MQL$ value of a specific road in each minute. The top 50 roads with the highest $MQL$ in the original scenario were selected. Due to space constraints, we place Figures 7, 8 and 9 in the appendix. In different scenarios, $\{MQL_t\}_{t=1,s1}^{30} \sim \{MQL_t\}_{t=1,s4}^{30}$ by MASDiff are more consistent with $\{MQL_t^*\}_{t=1,s1}^{30} \sim \{MQL_t^*\}_{t=1,s3}^{30}$ than other methods. This means that the learned policies can be used to conduct counterfactual reasoning in hypothetical scenarios.

### 4.4 RQ3: How is the scalability of the MASDiff framework

Scenario 1 is chose as the simulation environment. We increase the number of vehicles from 1,557 to 2,578 and 4,423 at time intervals of 0.5, 1, and 1.5 hours from 8:30 am. We adopt $MQL$ as the macro metric. The population size is also 500. The results are shown in Tables 1 and 2. In Table 2, running time denotes the time elapsed when the algorithm reaches convergence. The experimental results demonstrates that MASDiff significantly outperforms other methods both in performance and running time. The results obtained by LDM_CFG and LDM_PCFG are nearly identical to that from random sampling.

Table 1: $\|\{MQL_t^*\}_{t=1}^T - \{MQL_t\}_{t=1}^T\|$

| | 1557 | 2758 | 4423 |
|---|---|---|---|
| MASDiff | 1.69 | 2.19 | 2.46 |
| CGAN | 1.76 | 5.32 | 7.24 |
| LDM_IMIT | 2.02 | 2.40 | 2.91 |
| LDM_CFG | – | – | – |
| LDM_PCFG | – | – | – |

Table 2: Running time(minute)

| | 1557 | 2758 | 4423 |
|---|---|---|---|
| MASDiff | 101.25 | 139.17 | 218.33 |
| CGAN | 172.08 | 245.83 | 259.17 |
| LDM_IMIT | 113.75 | 123.33 | 140.83 |
| LDM_CFG | – | – | – |
| LDM_PCFG | – | – | – |

## 5 Conclusion

This paper tries to control the emergence of large scale multi-agent system. To solve this problem, the MASDiff framework are proposed. The experimental results verifies that MASDiff can achieve emergence control fairly well. To our best knowledge, MASDiff is the first framework targeting at controlling emergence in MAS. In future work, each part of the framework will be optimized.

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

# A APPENDIX

## A.1 IMPLEMENTATION DETAIL

This subsection describes the specific Markov process used in the experiment. Before giving the Markov Decision Process, we first introduce the topology of road network.

**Directed topology of a road network.** The directed topology of a road network is defined as $G = (V, E, X)$. $V$ represents the roads in the network, assume there are $m$ roads in $G$. $E$ denotes the connections between roads, assume there are $|E|$ connections. $X$ indicates traffic status for each road.

Figure 6(a) illustrates the physical road network. Figure 6(b) shows its corresponding topological graph $G = (V, E, X)$. $V = \{road_1, \ldots, road_{12}\}$, $E = \{road_7 \rightarrow road_4, road_7 \rightarrow road_2, \ldots, road_4 \rightarrow road_{11}\}$. $road_7 \rightarrow road_4$ indicates vehicles can travel from $road_7$ to $road_4$. $X = \{queue_{road_1}, \ldots, queue_{road_{12}}\}$ represents the traffic status of roads, for example queue length of each road. Actually, $G$ can be viewed as the complete MDP for learning $Q_\theta(x, a)$. We detail each component below.

**State $s$.** $s$ represents the current state, indicating the road and queue length of the vehicle, $s = (road, queue_{road})$.

**Action $a$.** $a$ is the navigation actions such as left turn, straight ahead, or right turn on the current road to the next road. For example, if the vehicle is in $road_4$, $a = (road_{10}, road_{11})$.

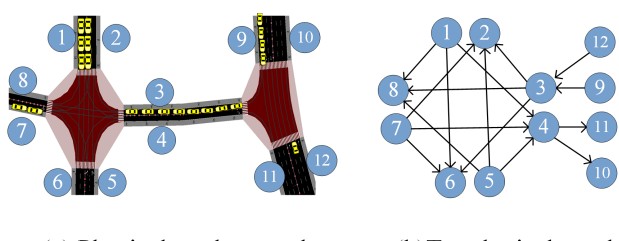

(a) Physical road network   (b)Topological graph

Figure 6: Topological graph of road network.

**Reward $r$.** $r$ is the reward obtained by steering to the next road at the current road.

**Policy $Q_\theta(s, a)$.** $Q_\theta(s, a)$ determines action $a$ for each state $s$. $Q_\theta(s, a)$ maximizes cumulative discounted rewards, such that $Q_\theta(s, a) = \text{Argmax}\left[\sum_{t=1}^T \gamma^t r(s_t, a_t)\right]$. $\gamma \in [0, 1)$ represents the discount factor.

For example, $((road_9, queue_{road_9}), road_3, (road_3, queue_{road_3}), r)$ is a piece of experience. It means that if vehicle in state $(road_9, queue_{road_9})$ takes action $road_3$, it will reach state $(road_3, queue_{road_3})$ and will get reward $r$. Let $\tau = \{(s_i, a_i, s_i')_t\}_{t=1}^T\}_{i=1}^{|E|}$ denote the offline trajectory for the vehicle. $|E|$ is the number of edges in $G$. Since the connection from road $s$ to $s'$ is fixed in a given traffic map, the number of experiences is exactly $|E|$.

However, if the local reward is not based on distance(such as "avoiding traffic lights" or "taking the favourite path"), reinforcement learning alone does not necessarily guarantee that the vehicle will reach the destination. This paper adopts a hybrid navigation policy of A* + Q-learning. For example, in Figure 6, when vehicle A travels from $road_9$ to $road_2$ using the A* alone. It selects the shortest path $road_9 \rightarrow road_3 \rightarrow road_2$. However, this path is the most congested. By combining A* with $Q_\theta(x, a)$, roads with longer queue lengths are assigned with lower rewards. The optimal path changes to a longer but less congested route: $road_9 \rightarrow road_11 \rightarrow ... \rightarrow road_5 \rightarrow road_2$.

**A* + Q-learning navigation policy.** The traditional A* navigation policy defines the total cost function as $r(s) = g(s) + h(s)$. $g(s)$ represents the actual cost from the starting point to the current node. $h(s)$ is the heuristic estimation function that estimates the potential cost to the endpoint. We designed $h(s)$ in Formula 12 by using an A* + Q-learning hybrid mechanism that not only meets routing requirements but also reflecting the vehicle's unique preferences.

$$h(s) = \|loc(s_{\text{current}}) - loc(s_{\text{end}})\|_2 - \lambda \cdot Q_\theta(s, a) \tag{12}$$

where $\|loc(s_{\text{current}}) - loc(s_{\text{end}})\|_2$ is the Manhattan distance between the current road of the vehicle and the destination of the vehicle. $\lambda$ is an adjustable hyper-parameter. $Q_\theta(s, a)$ is the preference policy learned by reinforcement learning. $s$ is the current state of the vehicle. $a$ is the navigation actions such as turning left to the next road. A* tries to minimize the cost of the path, while DQN tries to maximize the reward. So, we use the negative values of $Q_\theta(s, a)$ in Formula 12.

We employ offline Deep Q-learning (Kostrikov et al., 2022) to train $Q_\theta(x, a)$. The training process is not performed in real-time but completed once the vehicle first enters the road network by using the offline dataset.

A.2 EXPERIMENTAL RESULTS IN SCENARIO 2 AND SCENARIO 3

Figure 7~9 shows the results. The horizontal axis represents the minutes. The vertical axis represents the $MQL$ value of a specific road.

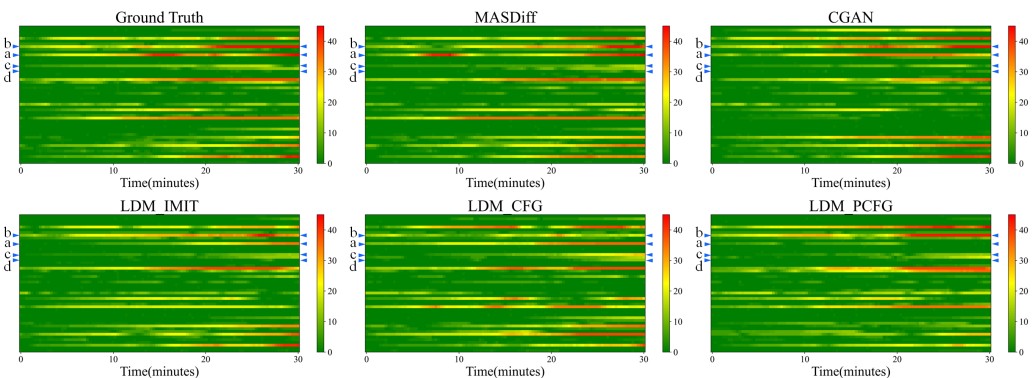

Figure 7: $\{MQL_t^*\}_{t=1}^{30}$ and $\{MQL_t\}_{t=1}^{30}$ of selected roads in scenario 1.

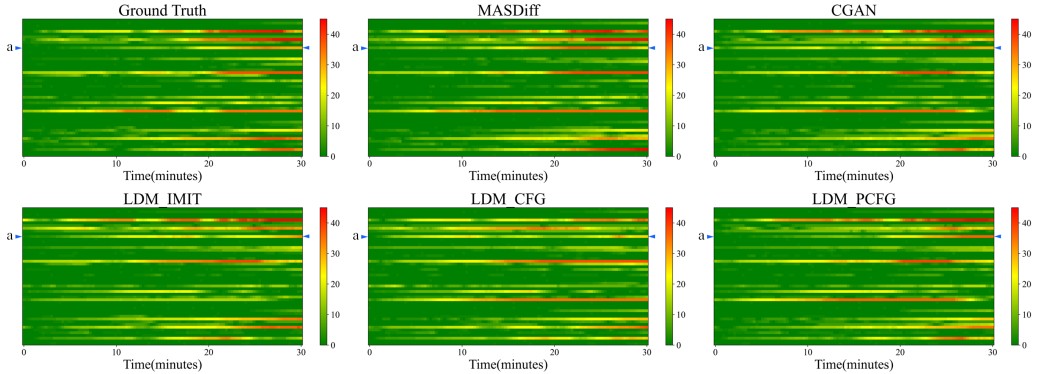

Figure 8: $\{MQL_t^*\}_{t=1}^{30}$ and $\{MQL_t\}_{t=1}^{30}$ of selected roads in scenario 2.

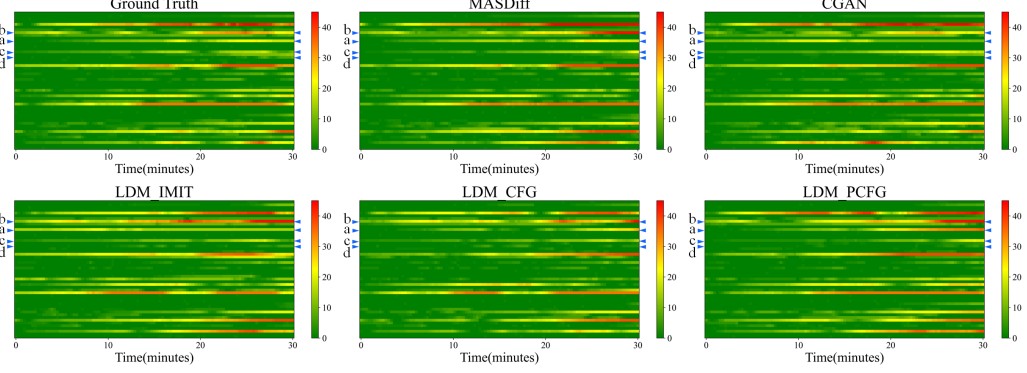

Figure 9: $\{MQL_t^*\}_{t=1}^{30}$ and $\{MQL_t\}_{t=1}^{30}$ of selected roads in scenario 3.

