# OpenReview forum: "MASDiff: Large Scale Multi-Agent System Emergence Control via Evolutionary Diffusion"
_ICLR.cc/2026/Conference — Submitted to ICLR 2026_

### Official Review · Reviewer_BhSg · 2025-10-23

**Soundness:** 1
**Presentation:** 1
**Contribution:** 1
**Rating:** 0
**Confidence:** 5

**Summary:**

This paper proposes **MASDiff**, a diffusion-model-based decentralized multi-agent reinforcement learning (MARL) framework aiming to control large-scale emergent behaviors in systems with over 1,000 agents.
The authors argue that existing CTDE (Centralized Training Decentralized Execution) and DTDE frameworks cannot effectively scale to such large systems.

To address this, **MASDiff** introduces:

* An **evolutionary diffusion model** that generates and optimizes sample trajectories instead of directly learning policies.
* An **offline RL component** that learns agent policies guided by diffusion-generated samples.
* A **dual-population evolution mechanism** combining policy and trajectory evolution for scalability.

Experiments are conducted in a **traffic signal control scenario** with over 1,500 simulated vehicles. The results suggest that the learned microscopic policies reproduce target macroscopic traffic patterns and allow limited counterfactual reasoning (“What if” scenarios).

**Strengths:**

* **Ambitious research goal:**
  The paper targets the challenging and important problem of controlling emergent behavior in *large-scale* multi-agent systems, which remains an underexplored area in MARL.

* **Creative methodological combination:**
  The integration of *diffusion models*, *evolutionary optimization*, and *offline reinforcement learning* is novel and conceptually appealing, bridging generative modeling with MARL.

* **Attention to scalability:**
  The dual-parallelism structure (policy and trajectory populations) is an interesting architectural idea that could, in principle, improve scalability to thousands of agents.

* **Structured methodology:**
  The derivation from an ideal oracle-based framework to the practical algorithm (Algorithm 2) is clearly explained and logically consistent.

* **Application relevance:**
  The traffic control domain provides a concrete and intuitive example to visualize emergent phenomena and control.

**Weaknesses:**

Despite its interesting premise, the paper falls short of ICLR standards in terms of **clarity, completeness, and experimental rigor**.

### (1) Lack of clear problem formulation

The introduction does not adequately explain what *core difficulty* arises when scaling MARL to thousands of agents, nor does it justify how the proposed method addresses it *in a generalizable way*.
The term “emergence control” is vaguely defined, with no formal or quantitative evaluation criterion.

### (2) Incomplete literature coverage

The related work section overlooks multiple important baselines and recent studies:

* In **online MARL**, key methods such as *RACE*[1], *SMPE*[2], and *Revisiting Off-policy MARL*[3] are not cited.
* In **diffusion-based RL**, critical algorithms like *EDP*[4], *IDQL*[5], *DPPO*[6], *OMAR*[7], *CFCQL*[8], *DOM2*[9], and *DoF*[10] are omitted.
  This incomplete coverage weakens both novelty and context.

[1]RACE: Improve Multi-Agent Reinforcement Learning with Representation Asymmetry and Collaborative Evolution. ICML 2023

[2]SMPE: Enhancing Cooperative Multi-Agent Reinforcement Learning with State Modelling and Adversarial Exploration. ICML 2025

[3]Revisiting Cooperative Off-Policy Multi-Agent Reinforcement Learning. ICML 2025

[4]EDP: Efficient diffusion policies for offline reinforcement learning. NeurIPS 2023

[5]IDQL: Implicit q-learning as an actor-critic method with diffusion policies.

[6]DPPO: Diffusion Policy Policy Optimization

[7]OMAR: Plan better amid conservatism: Offline multi-agent reinforcement learning with actor rectification. 2022 ICML

[8]CFCQL:Counterfactual Conservative Q Learning for Offline Multi-agent Reinforcement Learning. 2023 NeurIPS

[9]DOM2:Beyond Conservatism: Diffusion Policies in Offline Multi-agent Reinforcement Learning.

[10]DoF: A Diffusion Factorization Framework for Offline Multi-Agent Decision Making. ICLR 2025

### (3) Latent-space diffusion issues

The diffusion process is trained in a *latent space*, but the paper provides no analysis or ablation regarding how the encoder/decoder capacity affects model stability or performance.
This omission is crucial, as limited representation power could severely constrain trajectory generation and downstream policy learning.

### (4) Domain-specific and limited experiments

All evaluations are conducted on a **custom traffic control simulation**, without comparison on any **standard MARL benchmarks** (e.g., SMAC, MPE, Multi-Agent Mujoco).
Consequently, the claimed “general” scalability and emergence control ability lack empirical support.

### (5) Writing and presentation quality

The overall writing is **unpolished and difficult to follow**.
Many paragraphs repeat similar ideas, and key terms (“macro phenomenon”, “oracle”, “type III emergence”) are not precisely defined.
Figures and tables lack quantitative metrics that directly validate the claimed improvements.

### (6) Issues about the offline setting
I am curios about the issues of the offline setting and the algorithm seems hard to online finetuning or training from scratch based on the evoluational diffusion method. It seems a detailed explanation or experiments about offline-to-online and online setting.

While the paper presents a somewhat concept, it lacks the **conceptual clarity, comprehensive experimentation, and polished exposition** expected at ICLR. This paper is lack of novelty.
Substantial revisions, including clearer problem framing, expanded benchmark evaluations, and in-depth analysis of diffusion training, are needed before this work can be considered competitive.

**Questions:**

To improve the work, the authors are encouraged to address the following points:

1. **Generalization:**
   Can MASDiff be evaluated on standard MARL benchmarks (e.g., SMAC, MPE)?
   How does its performance scale with increasing agents under different task dynamics?

2. **Diffusion modeling details:**
   What is the dimensionality of the latent space? How sensitive are results to encoder architecture or noise schedule?

3. **Baseline coverage:**
   Why are methods like *Diffusion-QL*, *EDP*, *OMAR*, *IDQL*, etc., excluded from comparisons?

4. **Ablation studies:**
   What is the contribution of the *evolutionary mechanism* versus the diffusion model itself?

5. **Definition of emergence control:**
   How exactly is success measured? Is there a quantitative metric for macroscopic alignment?

6. **Reproducibility:**
   Will the authors release their simulation code, dataset, and hyperparameters for replication?

---

> ### Author Response · Authors · 2025-12-03
>
> Question 1:Generalization: Can MASDiff be evaluated on standard MARL benchmarks (e.g., SMAC, MPE)? How does its performance scale with increasing agents under different task dynamics?
>
> Answer to Q1: MASDiff be evaluated on standard MARL benchmarks. However, since the methods in the benchmark cannot solve the problem addressed in this paper, we omitted it from the comparison. The performance scale with increasing agents under different task dynamics are given in Table 1 and 2.
>
> Question 2: Diffusion modeling details: What is the dimensionality of the latent space? How sensitive are results to encoder architecture or noise schedule?
>
> Answer to Q2: The algorithm is not sensitive to the choice of encoder. We even incorporated a graph attention mechanism, but it did not lead to a significant improvement in performance. The evolutionary diffusion model is the core mechanism.
>
> Question 3: Baseline coverage: Why are methods like Diffusion-QL, EDP, OMAR, IDQL, etc., excluded from comparisons?
>
> Answer to Q3: Reward model learning methods are primarily divided into implicit reward modeling and explicit reward modeling. Implicit reward modeling aims to learn the intrinsic reward of each agent to facilitate better exploration, ultimately achieving the overall outcomes. Intrinsic reward shaping, self-supervised RL and credit assignment can be classified as implicit reward modeling. Explicit reward modeling aims to learn the behavioral preferences of agents by explicitly learning a reward model. The purpose is to ensure that the distribution of behavioral patterns derived from the reward model aligns with the statistical distribution observed in the expert data. Inverse RL, behavior cloning and Reinforcement Learning with Human Feedback(RLHF) can be classified as implicit reward modeling. Currently, there is hardly any method capable of simultaneously achieving the two aforementioned objectives, namely, learning explicit reward models to realize macro objectives in multi-agent systems. This is primarily because existing approaches are mainly designed to address conventional reinforcement learning problems, which do not require the integration of these two objectives. However, the problem addressed in this paper requires the integration of the two objectives, i.e. to learn the explicit reward model for each agent on the one hand and to archive the overall outcome.
>
> [1]RACE: Improve Multi-Agent Reinforcement Learning with Representation Asymmetry and Collaborative Evolution. ICML 2023
>
> This method operates on the parameters of policies. It cannot learn an explicit reward model for each individual agent.
>
> [2]SMPE: Enhancing Cooperative Multi-Agent Reinforcement Learning with State Modelling and Adversarial Exploration. ICML 2025
>
> The objective of this paper is to find an optimal joint policy which satisfies the optimal value function. It cannot learn an explicit reward model for each individual agent.
>
> [3]Revisiting Cooperative Off-Policy Multi-Agent Reinforcement Learning. ICML 2025
>
> This method cannot learn an explicit reward model for each individual agent.
>
> [4]EDP: Efficient diffusion policies for offline reinforcement learning. NeurIPS 2023
>
> [5]IDQL: Implicit q-learning as an actor-critic method with diffusion policies.
>
> [6]DPPO: Diffusion Policy Policy Optimization
>
> [7]OMAR: Plan better amid conservatism: Offline multi-agent reinforcement learning with actor rectification. 2022 ICML
>
> [8]CFCQL:Counterfactual Conservative Q Learning for Offline Multi-agent Reinforcement Learning. 2023 NeurIPS
>
> [9]DOM2:Beyond Conservatism: Diffusion Policies in Offline Multi-agent Reinforcement Learning.
>
> [10]DoF: A Diffusion Factorization Framework for Offline Multi-Agent Decision Making. ICLR 2025
>
> These methods all require offline training data while our method need not. However, for a practical reinforcement learning task, collecting training data is highly challenging. The method proposed in this paper overcomes this bottleneck.
>
>
>
> Question 4: Ablation studies: What is the contribution of the evolutionary mechanism versus the diffusion model itself?
>
> Answer to Q4：Since the traditional diffusion model can only generate data according to the given expert data, we introduce evolutionary diffusion model to generate data in the absence of any expert data. To steer the collaboration of all agents towards the specified macro-level objective, the macro-level objective is adopted as the fitness for the population. This mechanism transforms the multi-agent exploration based on intrinsic reward into the evolutionary exploration based on genetic operators. Moreover, the optimal individual retention strategy in the evolutionary diffusion model can address the non-stationary problem in MARL.
>
> Question 5: Definition of emergence control: How exactly is success measured? Is there a quantitative metric for macroscopic alignment?
>
> Answer to Q5：From line 85 to 92, formula 1 gives the formal  definition of emergence control.

---

### Official Review · Reviewer_dVau · 2025-10-28

**Soundness:** 3
**Presentation:** 3
**Contribution:** 2
**Rating:** 4
**Confidence:** 3

**Summary:**

The paper proposes MASDiff, a framework for controlling emergent behavior in large-scale multi-agent systems. Instead of directly optimizing policies, MASDiff learns to generate high-quality data samples—trajectories and rewards—that can lead to desired macroscopic outcomes. A conditional diffusion model is trained to approximate an oracle $O(\tau)$ mapping trajectories $\tau$ to optimal rewards $R$ that minimize macro-level error. An evolutionary sampling process (population-based selection, mutation via truncated diffusion, and denoising reconstruction) refines these samples iteratively. Agents then learn policies offline from the generated data using standard offline RL methods.

**Strengths:**

1. The design of the proposed Oracle is novel, which reframes large-scale MAS control as learning to generate trajectory–reward pairs rather than directly learning policies.
2. The diffusion-based oracle approximation is an intuitive way to model emergent control.
3. Experiments at the 1k–4k agent level within SUMO city network demonstrates good scalability.

**Weaknesses:**

1. Some of notations in this paper is confusing. For example, the $MQL$ appears for the first time in Line 414 without any explanation, which  makes section 5.4 hard to read.
2. Some experimental details need further clarification. The authors denote $LCC$ in Line 372 as the largest functional cluster of the traffic network. However, this definition is confusing for me. How is $LCC$ obtained? What kind of data is $LCC$? Is $LCC$ like the state sequences $X_{t=1}^T$?
3. Since this paper use the mse loss between the output of the proposed method and $LCC$ to evaluate the model, a correct and detailed definition of $LCC$ is important.
4. It seems the proposed MASDiff is similar to model-based RL. I suggest the authors to discuss the differences between MASDiff and model-based methods.
5. For evaluation, why didn't the autors use the training return curves? It would be good if the authors provide the learning curves for MASDiff and baselines.

I am willing to increase my score if the authors make some clarifications.

**Questions:**

See Weaknesses.

---

> ### Author Response · Authors · 2025-12-03
>
> Question 1: Some of notations in this paper is confusing. For example, the $MQL$ appears for the first time in Line 414 without any explanation, which makes section 5.4 hard to read.
>
> Answer to Q1: $MQL$ is adopted as the macro emergent traffic state which is the maximum queue length of a road segment in a certain period. The definition is given in he first paragraph of section 4.3.
>
> Question 2: Some experimental details need further clarification. The authors denote $LCC$ in Line 372 as the largest functional cluster of the traffic network. However, this definition is confusing for me. How is $LCC$ obtained? What kind of data is $LCC$? Is $LCC$ like the state sequences $X_{t=1}^T$?
>
> Question 3: Since this paper use the mse loss between the output of the proposed method and $LCC$ to evaluate the model, a correct and detailed definition of $LCC$ is important.
>
> Answer to Q2,Q3: The detail of $LCC$ is given by the reference paper "Guanwen Zeng, Jianxi Gao, Louis Shekhtman, Shengmin Guo, Weifeng Lv, Jianjun Wu, Hao Liu, Orr Levy, Daqing Li, Ziyou Gao, H. Eugene Stanley, and Shlomo Havlin. Multiple metastable
> network states in urban traffic. Proceedings of the National Academy of Sciences, 117(30):17528–17534, 2020. doi: 10.1073/pnas.1907493117. URL https://www.pnas.org/doi/abs/10.1073/pnas.1907493117.".
>
> Question 4:  It seems the proposed MASDiff is similar to model-based RL. I suggest the authors to discuss the differences between MASDiff and model-based methods.
>
> Answer to Q4: MASDiff is completely different with model-based methods.The purpose and the methods of this paper is as follows:
>
> Reward model learning methods are primarily divided into implicit reward modeling and explicit reward modeling. Implicit reward modeling aims to learn the intrinsic reward of each agent to facilitate better exploration, ultimately achieving the overall outcomes. Intrinsic reward shaping, self-supervised RL and credit assignment can be classified as implicit reward modeling. Explicit reward modeling aims to learn the behavioral preferences of agents by explicitly learning a reward model. The purpose is to ensure that the distribution of behavioral patterns derived from the reward model aligns with the statistical distribution observed in the expert data. Inverse RL, behavior cloning and Reinforcement Learning with Human Feedback(RLHF) can be classified as implicit reward modeling. Currently, there is hardly any method capable of simultaneously achieving the two aforementioned objectives, namely, learning explicit reward models to realize macro objectives in multi-agent systems. This is primarily because existing approaches are mainly designed to address conventional reinforcement learning problems, which do not require the integration of these two objectives. However, the problem addressed in this paper requires the integration of the two objectives, i.e. to learn the explicit reward model for each agent on the one hand and to archive the overall outcome.
>
> Question 5: For evaluation, why didn't the autors use the training return curves? It would be good if the authors provide the learning curves for MASDiff and baselines.
>
> Answer to Q5：Since the exploration mechanism serves as the core component of this paper, the experimental performance improves through evolution. Typically, evolutionary computation presents the evolutionary curve of the optimal results.

---

### Official Review · Reviewer_qWsX · 2025-10-29

**Soundness:** 2
**Presentation:** 2
**Contribution:** 2
**Rating:** 2
**Confidence:** 3

**Summary:**

This paper introduces MASDiff, a novel framework for controlling emergent macroscopic phenomena in large-scale multi-agent systems (with over 1000 agents). The core challenge addressed is that traditional multi-agent reinforcement learning (MARL) frameworks like CTDE and DTDE struggle at this scale. The key contribution is a paradigm shift from searching in the policy space to searching in the space of training data. MASDiff uses a conditional diffusion model, guided by an evolutionary algorithm, to generate optimal reward and trajectory data. Individual agent policies are then trained using offline RL on this generated data. The framework is validated in a large-scale traffic simulation, demonstrating its ability to steer the system towards a target macroscopic state and perform counterfactual reasoning.

**Strengths:**

- The paper tackles a highly significant and challenging problem: controlling emergence in massive multi-agent systems. The core idea is clear. Shifting the search from the high-dimensional, unstable space of policy parameters to the manifold of "good" training data is an approach that sidesteps online large-scale MARL.

- The proposed MASDiff framework is interesting, as it coherently integrates diffusion models for generative search, evolutionary strategies for exploration, and offline RL for policy learning. The "Actor-Diffusion-Learner" architecture is a clean and scalable conceptualization.

- The experimental validation is another strong point. The use of a realistic, large-scale traffic simulator (SUMO) with real-world map data lends credibility to the applicability of this work. The experiments are designed to answer specific research questions, and the demonstration of counterfactual reasoning (RQ2) is a particularly compelling showcase of the framework's potential utility beyond simple pattern matching.

**Weaknesses:**

1.  The paper only compares MASDiff against its own ablations. The central claim is that existing CTDE / DTDE MARL frameworks "can hardly work" at this scale, but this claim is not experimentally substantiated. Without comparisons to established, state-of-the-art MARL algorithms (e.g., MAPPO, QMIX), it is very difficult to gauge the performance and efficiency gains of MASDiff.

2.  Algorithm 2 presents a very high-level loop that is hard to follow, with multiple simulation, update, and evolution steps. A more detailed diagram illustrating the flow of data between the policy population, the sample population, and the diffusion model could improve understanding.

3.  The proof for Proposition 1 is hand-wavy; equating this complex iterative process with "classical policy iteration" needs a more rigorous justification and references.

4.  The framework has many associated hyperparameters (e.g., population size `M=500`, diffusion steps `t`, scaling factor `γ`). The paper provides no discussion on the sensitivity to these parameters or the process for tuning them. For a framework this complex, understanding sensitivity is crucial for reproducibility and practical application.

**Questions:**

1.  The goal of this paper is a bit confusing, it might be good to discuss the relation of this work to inverse reinforcement learning?

2. Could you elaborate on the choice of baselines? The main claim is that existing MARL frameworks are insufficient, but no such frameworks were included in the comparison. Why not compare against adapted versions of established algorithms like MAPPO or PBT, or with inverse RL?

3.  The framework has a large number of hyperparameters (population size, evolutionary parameters, diffusion steps, etc.). Could you provide some insight into their tuning? Is the framework's performance highly sensitive to these choices, for instance, the population size `M`?

4. The methodology appears to be quite related to mean field methods such as mean field games and mean field control [1-3], which describes the macroscopic behavior resulting from microscopic agent behavior and was also similarly applied to traffic control [4]. Is there potential for a synthesis of both worlds?

[1] Perrin, S., et al. Mean Field Games Flock! The Reinforcement Learning Way. IJCAI (2021).

[2] Carmona, R. et al. Model-free mean-field reinforcement learning: mean-field MDP and mean-field Q-learning. The Annals of Applied Probability 33.6B (2023): 5334-5381.

[3] Cui, K., et al. Learning Decentralized Partially Observable Mean Field Control for Artificial Collective Behavior. ICLR (2024).

[4] Wu, M., et al. Participatory traffic control: Leveraging connected and automated vehicles to enhance network efficiency. Transportation Research Part C: Emerging Technologies 166 (2024): 104757.

---

> ### Author Response · Authors · 2025-12-03
>
> Questions 1、The goal of this paper is a bit confusing, it might be good to discuss the relation of this work to inverse reinforcement learning?
>
> Questions 2、Could you elaborate on the choice of baselines? The main claim is that existing MARL frameworks are insufficient, but no such frameworks were included in the comparison. Why not compare against adapted versions of established algorithms like MAPPO or PBT, or with inverse RL?
>
> Answer to Q1,Q2: Reward model learning methods are primarily divided into implicit reward modeling and explicit reward modeling. Implicit reward modeling aims to learn the intrinsic reward of each agent to facilitate better exploration, ultimately achieving the overall outcomes. Intrinsic reward shaping, self-supervised RL and credit assignment can be classified as implicit reward modeling. Explicit reward modeling aims to learn the behavioral preferences of agents by explicitly learning a reward model. The purpose is to ensure that the distribution of behavioral patterns derived from the reward model aligns with the statistical distribution observed in the expert data. Inverse RL(Wang et al., 2024), behavior cloning(Zhou et al., 2024) and Reinforcement Learning with Human Feedback(RLHF)(Casper et al., 2023) can be classified as implicit reward modeling. Currently, there is hardly any method capable of simultaneously achieving the two aforementioned objectives, namely, learning explicit reward models to realize macro objectives in multi-agent systems. This is primarily because existing approaches are mainly designed to address conventional reinforcement learning problems, which do not require the integration of these two objectives. However, the problem addressed in this paper requires the integration of the two objectives, i.e. to learn the explicit reward model for each agent on the one hand and to archive the overall outcome.
>
> Questions 3:The framework has a large number of hyperparameters (population size, evolutionary parameters, diffusion steps, etc.). Could you provide some insight into their tuning? Is the framework's performance highly sensitive to these choices, for instance, the population size M?
>
> Answer to Q3: The framework's performance  is not quite sensitive to the choice of hyperparameters.Within the commonly used range of the parameter, the algorithm's performance varies little.
>
>
> Questions 4:The methodology appears to be quite related to mean field methods such as mean field games and mean field control [1-3], which describes the macroscopic behavior resulting from microscopic agent behavior and was also similarly applied to traffic control [4]. Is there potential for a synthesis of both worlds?
>
> Answer to Q4:Mean Field Reinforcement Learning requires the agents in the swarm are interchangeable. This frameworks can not solve the emergence control problem.

---

### Meta-Review · Area_Chair_BewT · 2025-12-21

**Summary:**

The paper proposes MASDiff, a method for large-scale multi-agent systems. MASDiff learns to generate high-quality data samples that can lead to desired outcomes. It then adopts a diffusion model to model data samples. An evolutionary process is used to refine the diffusion model.

The strengths of this work are summarized as follows.
1. The research topic is timely and important.
2. The proposed combination of the diffusion model and the evolutionary process is interesting.
3. The large-scale experimental results are promising.

The weaknesses of this work are listed as follows.
1. Insufficient related work coverage.
2. Multiple diffusion-based methods and MARL methods are not adopted as the baselines.
3. The writing of this work should be improved.

**Reviewer Concerns:**

The reviewers concern with the problem formulation, related work, incomplete baselines, and the writing.

**Reviewer Scores:**

The reviewers will not change the review scores.

---

### Decision · Program_Chairs · 2026-01-26

Reject